# Effectiveness of Vascular Catheter Removal Versus Retention in Non-ICU Patients with CRBSI or CABSI in Retrospective, Single-Center Study

**DOI:** 10.3390/microorganisms13051085

**Published:** 2025-05-07

**Authors:** Giovanni De Capitani, Marta Colaneri, Claudia Conflitti, Fabio Borgonovo, Lucia Galli, Giovanni Scaglione, Camilla Genovese, Rebecca Fattore, Monica Schiavini, Beatrice Caloni, Daniele Zizzo, Nicola Busatto, Antonio Gidaro, Alba Taino, Maria Calloni, Francesco Casella, Arianna Bartoli, Chiara Cogliati, Emanuele Palomba, Spinello Antinori, Andrea Gori, Antonella Foschi

**Affiliations:** 1Department of Infectious Diseases, Luigi Sacco Hospital, ASST Fatebenefratelli Sacco, 20157 Milan, Italymarta.colaneri@unimi.it (M.C.); borgonovo.fabio@asst-fbf-sacco.it (F.B.); scaglione.giovanni@asst-fbf-sacco.it (G.S.); genovese.camilla@asst-fbf-sacco.it (C.G.); fattore.rebecca@asst-fbf-sacco.it (R.F.); schiavini.monica@asst-fbf-sacco.it (M.S.); caloni.beatrice@asst-fbf-sacco.it (B.C.); zizzo.daniele@asst-fbf-sacco.it (D.Z.); busatto.nicola@asst-fbf-sacco.it (N.B.); spinello.antinori@unimi.it (S.A.); andrea.gori@unimi.it (A.G.); foschi.antonella@asst-fbf-sacco.it (A.F.); 2Department of Biomedical and Clinical Sciences, Università degli Studi di Milano, 20157 Milan, Italy; cogliati.chiara@asst-fbf-sacco.it; 3National PhD Programme in One Health Approaches to Infectious Diseases and Life Science Research, Department of Public Health, Experimental and Forensic Medicine, University of Pavia, 27100 Pavia, Italy; claudia.conflitti01@universitadipavia.it; 4Department of Internal Medicine, Luigi Sacco Hospital, ASST Fatebenefratelli Sacco, 20157 Milan, Italy; gidaro.antonio@asst-fbf-sacco.it (A.G.); taino.alba@asst-fbf-sacco.it (A.T.); calloni.maria@asst-fbf-sacco.it (M.C.); casella.francesco@asst-fbf-sacco.it (F.C.); bartoli.arianna@asst-fbf-sacco.it (A.B.)

**Keywords:** catheter-related bloodstream infection (CRBSI), catheter-associated bloodstream infection (CABSI), vascular access device (VAD), removal, retention

## Abstract

Catheter-associated bloodstream infections (CABSIs) and catheter-related bloodstream infections (CRBSIs) are significant causes of morbidity and mortality worldwide. The current practice favors the removal of vascular access devices (VADs); however, the evidence on this topic remains inconclusive. This study evaluates the clinical outcomes in terms of in-hospital mortality and catheter retention vs. removal in CABSI and CRBSI cases. A retrospective, observational, single-center study was conducted at Luigi Sacco Hospital, Milan, Italy (May 2021–December 2023), and it analyzed non-ICU adult patients with VADs diagnosed with CRBSIs or CABSIs. Clinical and microbiological data were collected to assess the outcomes based on catheter management. Among 1874 patients with VADs, 147 were included, with 164 VAD infection events (92 CABSIs and 72 CRBSIs). Overall, 35 (23.8%) patients with CABSIs and CRBSIs died. Out of those who retained the catheter 19 (35.8%) patients died, while among removal patients 16 (17%) died (*p* = 0.018). A *Candida* spp. isolation was found to be significantly associated with a higher likelihood of catheter removal (*p* = 0.04). Our findings suggest that, in non-ICU CRBSI and CABSI cases, VAD removal may be associated with improved outcomes when feasible.

## 1. Introduction

The placement of a vascular access device (VAD) is a very common practice, with approximately 75% of patients undergoing a VAD insertion during hospitalization [1]. VADs include peripheral devices, like short peripheral cannulas (<6 cm), long peripheral catheters (6–15 cm), and midline catheters (20–25 cm), as well as central VADs (CVADs) used for chemotherapy, fluids, antibiotics, and nutrition. CVADs are further classified as peripherally inserted central catheters (PICCs), femoral inserted central catheters (FICCs), and centrally inserted central catheters (CICCs) [2].

VADs can lead to both local complications, such as device dislocations, phlebitis, insertion site infections, and catheter colonization, and systemic complications, including venous thrombosis and infections. Among these, catheter-related bloodstream infections (CRBSIs) and catheter-associated bloodstream infections (CABSIs) are particularly frequent healthcare-associated infections (HAIs) [3]. These infections represent a major cause of a prolonged length of stay (LOS), increased morbidity and mortality, and higher hospitalization costs [4].

Most studies focusing on CRBSIs and CABSIs in intensive care units (ICUs) reported an incidence ranging from 0.18 to 3.84 per 1000 catheter-days [5,6]. Although the incidence of CRBSIs and CABSIs significantly increased outside ICU settings [7], data on their impact in non-ICU patients remain limited.

Moreover, current guidelines on CRBSIs and CABSIs diagnosis and management are controversial [8], and one of the main daily clinical challenges remains whether to remove or retain the catheter once an infection has been diagnosed. According to some authors [9], several specific situations require prompt VAD removal. These include septic shock, endocarditis, septic thrombophlebitis, and insertion site infections. Moreover, the persistence of positive blood cultures (BCs) at 72 h after an effective antimicrobial therapy administration and the infections caused by *Staphylococcus aureus*, *Candida* spp., or non-tuberculous mycobacteria also require VAD removal. Otherwise, catheter retention might be attempted in some non-complicated patients whose infection is caused by coagulase-negative *Staphylococci* (CoNS), *Corynebacterium* spp. (except *Corynebacterium jeikeium*), and Gram-negative bacteria [9]. Other authors also showed a higher mortality of patients with VAD retention in cases of infections with multi-drug-resistant organisms (MDROs) [10].

Following VAD removal, systemic antimicrobial therapy is generally administrated, while in the case of VAD retention, guidelines recommend combining systemic antimicrobials with lock therapy [11].

The lack of clear indications often leads physicians to make clinical decisions based on their personal experience [12,13]. Better management can be achieved through the establishment of specific vascular access teams (VATs) that can guide clinicians in the prevention, early diagnosis, and management of VAD complications, including whether to remove or retain the VAD [13,14,15,16].

This study investigated the clinical outcomes in terms of the LOS, ICU admission, and in-hospital mortality of CRBSI and CABSI patients with the removal or retention of VADs. Additionally, the clinical and microbiological characteristics associated with the clinicians’ propensity to remove or retain the catheter in these patients were assessed.

## 2. Materials and Methods

### 2.1. Study Design and Clinical Setting

This retrospective, monocentric, observational study was conducted at Luigi Sacco Hospital from 1 May 2021 to 1 December 2023.

The Luigi Sacco Hospital has had an established VAT since 2018, which provided standardized procedures and data reporting. All devices were positioned following the protocol “Safe insertion of PICCs (SIP)” [17].

### 2.2. Study Population

At the Luigi Sacco University Hospital in Milan, the Department of Infectious Diseases maintained a systematic registry of all VADs inserted by the VAT in standard wards, excluding the ICU, between 1 May 2021 and 1 December 2023. Exclusion criteria were as follows: catheter-length under 15 cm (short peripheral cannulas and long peripheral catheters), patients under 18 years of age, pregnant women, and those whose diagnosis occurred while hospitalized in the ICU or within 48 h of being transferred to a non-ICU ward.

From this dataset of VAD insertions, patients who were diagnosed with CRBSIs and CABSIs were selected. All patients were followed until hospital discharge, transfer to ICU, or in-hospital death.

### 2.3. Ethics

All subjects provided their written informed consent for inclusion. This study was conducted according to the Declaration of Helsinki, and the protocol was approved by the Luigi Sacco Hospital Institutional Review Board (Research Ethics Committee approval number 30236/2024).

### 2.4. Definitions

CRBSI was defined by the presence of at least one of the following criteria: (i) differential time to positivity (DTP), namely the isolation of the same microorganism in BCs drawn simultaneously from a peripheral vein and VAD, where VAD cultures become positive at least two hours before peripheral vein cultures, and the (ii) isolation of the same microorganism from the catheter’s tip and BCs drawn from a peripheral vein.

The definitions of CABSI and central-line-associated bloodstream infection (CLABSI) used in this study were based on the 2024 Infusion Nursing Society Standards of Practice and the Centers for Disease Control and Prevention’s National Healthcare Safety Network [18,19]. CABSI referred to bloodstream infections (BSIs) originating from either peripheral and/or central VADs, excluding infections in other sites. CLABSI refers to BSIs originating from CVADs that are unrelated to an infection at another site.

### 2.5. Available Data

Available data included patients’ demographics (biological sex and age), comorbidities (namely cardiovascular, endocrine, oncological, autoimmune, and chronic kidney and liver diseases), and history of CRBSIs. VAD characteristics comprised type of catheter, number of lumens, insertion site, and number of attempts.

Clinical information, like retention or removal of VAD, and administration of systemic and/or lock therapy were collected. Data about LOS, time before VAD infection onset (time to infection, TTI), and total VAD dwell time were also included.

Additionally, microbiological data including surveillance nasal and rectal swabs, BCs, and antimicrobial susceptibility testing (AST) were assessed. Rectal swabs were processed on selective chromogenic media (CHROMID^®^ CARBA, CHROMID^®^ ESBL, CHROMID^®^ VRE bioMérieux, Marcy-l’Etoile, France) and incubated at 35–37 °C for 18–48 h. Colonies identification was performed by matrix-assisted laser desorption ionization time-of-flight mass spectrometry (MALDI-TOF MS) (Bruker Daltonics, Bremen, Germany) and analyzed using the BioTyper version 3.1. Carbapenemase production was confirmed using the NG-Test^®^ CARBA-5 immunoassay (NG Biotech Laboratories, Guipry, France) or molecular-based testing (Xpert Carba-R kit, Cepheid, Sunnyvale, CA, USA). The first isolate was subjected to antimicrobial susceptibility testing (AST) using the BD Phoenix 50™ instrument (Becton Dickinson, Franklin Lakes, NJ, USA). MIC values were interpreted according to the European Committee on Antimicrobial Susceptibility Testing (EUCAST). BCs were incubated into the BD BACTEC FX system (Becton Dickinson, Franklin Lakes, NJ, USA), and positive blood culture bottles were subjected to Gram staining, subculturing, species identification, and antimicrobial susceptibility testing (AST) according to laboratory procedures.

### 2.6. Objectives

The primary objective of this study was to evaluate the clinical outcomes of catheter retention vs. removal in patients with CRBSI and CABSI, specifically in terms of all-cause in-hospital mortality, ICU admission, and LOS.

The secondary objectives were as follows:To analyze the microbiological and clinical characteristics of patients with CRBSI and CABSI, comparing those who underwent catheter removal with those who did not.To identify the factors associated with catheter removal in cases of CRBSI and CABSI.

### 2.7. Statistical Analysis

Continuous variables were expressed as medians (interquartile range [IQR]) and categorical variables as counts and percentages. Differences between CRBSI and CABSI groups (catheter removal vs. catheter retention) were tested with the Mann–Whitney test for continuous variables and Chi-square test for categorical variables. Mixed effect logistic regression was applied to assess the variables associated with catheter removal; the following variables were selected by the VAT based on clinical criteria: age; biological sex; number of comorbidities; positive rectal swab; polymicrobial infection; and identification of *Candida* spp., *Staphylococcus aureus*, or CoNS as causative agents of the infection. Data were analyzed using R software, version 4.3.3, and statistical significance was accepted at the 5% level.

## 3. Results

The dataset for this study included data on 1874 patients who underwent a total of 2244 VAD placements. Among them, 147 patients were diagnosed with a CRBSI or CABSI. Since some patients experienced more than one VAD placement and infectious event, the total number of infections was 164. Specifically, 63 patients had a CRBSI, with 17 retaining and 46 removing the VAD. Additionally, 84 patients had a CABSI, with 36 retaining and 48 removing the VAD.

The median age of CRBSI patients was significantly higher in the retention group compared to the removal (83 years [78–89] vs. 72 years [57.2–79.8], *p* = 0.003). For CABSI patients, no significant differences were observed between the two groups (80 years [72–87.2] vs. 74 years [65.2–82.2], *p* = 0.110).

The median number of comorbidities was similarly distributed between the retention and removal patients in both the CRBSI and CABSI groups (3 [2, 4] vs. 3 [1, 4], *p* = 0.401 and 2.5 [1.8, 4] vs. 3 [2, 3], *p* = 0.607, respectively).

Ongoing antibiotic therapy at the time of infection was reported in 11 (64.7%) CRBSI patients in the retention and 16 (34.8%) in the removal group (*p* = 0.065) and 16 (45.7%) CABSI patients in the retention and 23 (47.9%) in the removal group (*p* = 1.000). The classes of antibiotics used were equally distributed, with penicillins being the most common (27 patients overall, 18.4% of total patients).

The median LOS in the CRBSI group was shorter in the patients in the retention compared to the removal group (32 days [26–49] vs. 46.5 days [32.5–66.8], *p* = 0.0386), while no significant differences were noted in the CABSI group (34 days [19–56.5] vs. 39.5 days [28.5–65], *p* = 0.250).

The median VAD dwell time in the CRBSI group was 15 days [8–17] in the retention group and 12 [7.0–21.8] in the removal group (*p =* 0.828), while in the CABSI patients it was 8 days [4.8–13] for the retention group and 11 [5.0–20.2] for the removal one (*p* = 0.121).

The median TTI in the CRBSI group was 25 days (19–32.5) among retention patients and 19 days (10–24.8) among the removal group (*p* = 0.173); in the CABSI group it was 20 (11.8–29.2) and 14 (7.8–24) days, respectively (*p* = 0.093).

Overall, 35 (23.8%) patients with CABSIs and CRBSIs died. Out of those who retained the catheter 19 (35.8%) patients died, while among retention patients 16 (17%) died (*p* = 0.018). In the CABSI patients, a significantly higher number of deaths was observed in the retention group (13, 36.1% vs. 6, 12.5%, *p* = 0.022). Among CRBSI patients, 6 deaths (35.3%) occurred in the retention group compared to 10 in the removal group (21.7%, *p* = 0.441) (Table 1).

While the PICC was the most used catheter in the CRBSI group (39 placements, 54.2%), the midline was the most inserted VAD in the CABSI group (61 placements, 66.3%). Overall, the most common insertion site was the thigh, with 78 placements (46.6%), while in the CRBSIs group the arm was the most used, with 40 placements (55.6%) (Table 2).

Epidemiological characteristics, microbiological profiles, and management strategies stratified by catheter types are reported in Appendix A. A higher prevalence of CRBSIs was observed among PICCs (65.0%), while among midlines the CABSI was the most prevalent type of infection (70.1%). When considering the management of infections originating from CICCs, five out of six cases (83.3%) underwent catheter removal and re-positioning under 48 h. However, the limited sample size prevented us from making relevant observations on individual microbiological isolates.

When considering nasal and rectal swabs, no significant difference was observed between the retention and removal groups. Rectal swabs aimed to identify MDROs, including vancomycin-resistant enterococci *(VRE),* extended beta-lactamase-producing *Escherichia coli*, *Pseudomonas aeruginosa*, *Staphylococcus aureus*, *Acinetobacter baumannii*, *Candida* spp., and New Delhi metallo-beta-lactamase-producing *Klebsiella pneumoniae* (Table 1).

When comparing differences in BC isolates between groups, it was found that among CABSI patients, Enterobacteriaceae isolates were more frequent in the retention group (10, 27.8% vs. 3, 6.2%, *p* = 0.017) (Table 1).

Considering all 164 VAD placements, CoNS was the most frequent microbiological isolate in both the CRBSI and CABSI groups. In the CRBSI group, a significantly higher number of *Candida* spp. isolates was observed (29.2% of CRBSIs vs. 1.1% of CABSIs, *p* < 0.001) as well as polymicrobial infections (22.2% of CRBSIs vs. 9.8% of CABSIs, *p* = 0.047) (Table 2).

BCs at 72 h were performed in only 42 patients overall (28.6%), more frequently in those who underwent VAD removal rather than retention (35.1% vs. 17%, *p* = 0.031). BSIs persisted in six cases (18.2%), all of which belonged to the removal group (Table 3).

The most common clinical decision in the case of CRBSIs was to remove the catheter and replace it within 48 h (26 cases, 30.1%). In the CABSI group, catheter retention with a simultaneous systemic antibiotic therapy was the most common practice, significantly more frequently than in CRBSIs (32, 34.8% vs. 11, 15.3% *p* < 0.008) (Table 2).

The multivariable mixed effect logistic analysis only identified a significant association of a *Candida* spp. isolation to the propensity for the removal of the infected VAD (OR 12.2, 95% CI 1.0–146.5). No other characteristics were found to be associated with the decision to remove the VAD (Table 4).

## 4. Discussion

Our study revealed a higher in-hospital mortality rate among patients with both catheter-associated bloodstream infections (CABSIs) and catheter-related bloodstream infections (CRBSIs) in the retention group, which is similar to findings in the CABSI group alone. However, when we focused solely on CRBSIs, we did not reach statistical significance, even though a higher percentage of patients who retained the vascular access device (VAD) died compared to those who had it removed (35.3% vs. 21.7%).

The observed discrepancy in mortality patterns between CRBSIs and CABSIs may be partially explained by the small sample of CRBSIs and by the microbiological profiles. In our study, in fact, we recorded a considerable number of *Candida* spp. infections (19); the majority of which (16) were in the CRBSI removal group. Since a Candida spp isolation is associated with more severe clinical outcomes regardless of the decision to retain or remove the catheter [20], this may have contributed to the increased mortality rate observed in the CRBSI removal group.

Our findings suggest a potential association between VAD retention and adverse outcomes, aligning with previous studies that show better outcomes when the catheter is promptly removed [21]. In this retrospective cohort of 430 individuals with CABSIs caused by several MDROs, catheter retention was strongly associated with a 30-day all-cause mortality.

On the other hand, some authors suggested that, when specifically dealing with *Enterococcus* spp. CABSIs, catheter retention might be as effective as removal [22]. However, it is important to note that these studies focused exclusively on ICU patients with CVC-related infections, leaving a significant gap in the literature regarding non-ICU settings and non-CVC VADs.

We found that the median LOS was 32 (26–49) and 46.5 days (32.5–66.8) in the CRBSI retention and removal group, while among CABSIs it was 34 (19–56.5) and 39.5 (28.5–65) days, respectively. These numbers are found to be lower than other findings on CRBSI and CABSI cases in ICUs, where the median LOS ranged from 20.8 to 69.1 days [23].

Among CRBSI patients, those whose catheter was removed were found to have a significantly longer LOS. This finding may be explained by the fact that patients who underwent catheter removal were more likely to be in septic shock or have infections caused by pathogens, such as *Candida* spp. and *Staphylococcus aureus*, which may require longer treatments [24,25,26].

The median TTI observed among CRBSIs was 25 days (19–32.5) among the retention group and 19 days (10–24.8) among the removal one, while in the CABSI group it was 20 (11.8–29.2) and 14 days (7.8–24). These numbers were found to be in line with what can be found in the literature among patients who underwent PICC placements and developed CABSIs or CRBSIs [27].

Regarding the clinical characteristics of patients, a similar distribution of comorbidities was observed among groups, indicating a homogeneous sample. Among CABSIs, a significantly older median age was observed in the retention group compared to the removal group. This result suggests a possible propensity toward a more conservative approach in elderly patients, favoring catheter retention rather than removal and avoiding the potential need for a subsequent invasive procedure to re-position a new vascular access.

The microbiological analyses revealed that CoNS were the most frequent pathogens involved in both CRBSIs and CABSIs, which is consistent with the previous literature [28]. The isolation of Enterobacteriaceae in CABSI patients was more frequent in the retention group. This may be explained by the fact that, at the time of infection, the treating physicians may not have been fully convinced that the catheter was the primary source of the bloodstream infection, since these microorganisms are frequent causes of urinary tract infections, and consequently felt less compelled to remove the catheter.

An interesting finding pertains to BCs at 72 h. Repeating BCs is essential for managing catheter-related infections, as positivity at 72 h indicates the need for catheter removal [9]. Despite this, only 28.6% of patients in our study underwent BCs at 72 h, with an even lower rate observed in those who had their catheter retained and would thus require removal in the case of persistent BSIs. This finding prevented us from making any significant conclusions about the persistence of bloodstream infections; nonetheless, it underscores the need for more standardized protocols and educational initiatives led by VATs to ensure the optimal monitoring of infections

We also found that a *Candida* spp. infection was associated with catheter removal. This is consistent with the scientific literature, which considers *Candida* spp. positive BCs as a determining factor for the need to remove the catheter [9].

Some limitations must be acknowledged. Firstly, this study’s retrospective and single-centered nature prevented the establishment of a direct causal relationship between catheter management and clinical outcomes and the generalizability of our conclusions. Moreover, in some cases the data could not be retrieved because they were either not collected at the time or not documented in medical records. Furthermore, our study may indeed be affected by confounding by indication, as the decision to remove or retain the catheter made by the clinician could have been influenced by unmeasured clinical factors or by the perceived severity of the patient’s condition at the specific time of the infection. Another limitation is represented by the fact that we only considered all-cause mortality, so not all deaths can be attributable to CRBSIs and CABSIs. Lastly, the relatively small number of infections analyzed did not allow for an in-depth stratified examination of some subgroups, particularly regarding microbiological isolates and different VAD types.

However, it is important to highlight that we had access to a comprehensive dataset containing detailed information on all analyzed patients. This represents a key strength of our study, as it allows for a thorough evaluation of VAD management. Additionally, the presence of a VAT further reinforces the reliability of our findings. Most of the studies available in the literature focus on ICU patients, whereas our study provides valuable insights into a non-ICU population, offering a broader perspective on VAD management in different clinical settings.

## 5. Conclusions

Our findings suggest that in non-ICU patients with CRBSIs or CABSIs, mortality was higher among those whose VAD was retained. To better understand this association, prospective studies exploring the impact of early catheter removal would be beneficial.

## Figures and Tables

**Table 1 microorganisms-13-01085-t001:** Epidemiological and clinical characteristics of patients with CRBSI or CABSI.

		CRBSI	CABSI				
		Retention	Removal	*p*	Retention	Removal	*p*
Biological sex	Male	6 (35.3)	22 (47.8)		17 (47.2)	24 (50.0)	
Age (years)	Median [IQR]	83 [78, 89]	72 [57.2, 79.8]	0.003	80 [72.0, 86.2]	74 [65.2, 82.2]	0.110
Diabetes mellitus		4 (23.5)	14 (30.4)	0.822	9 (25.0)	9 (18.8)	0.672
Cardiovascular diseases		11 (64.7)	15 (32.6)	0.044	10 (27.8)	20 (41.7)	0.278
Pulmonary diseases		5 (29.4)	10 (21.7)	0.763	5 (13.9)	11 (22.9)	0.446
Neurological diseases		10 (58.8)	18 (39.1)	0.266	9 (25.0)	22 (45.8)	0.083
Chronic liver disease		0 (0.0)	3 (6.5)	0.679	1 (2.8)	3 (6.2)	0.824
Chronic kidney disease		2 (11.8)	5 (10.9)	1.000	4 (11.1)	4 (8.3)	0.957
Autoimmune diseases		1 (5.9)	6 (13.0)	0.725	4 (11.1)	10 (20.8)	0.374
Solid neoplasia		6 (35.3)	14 (30.4)	0.949	6 (16.7)	3 (6.2)	0.241
Oncohematologic diseases		0 (0.0)	1 (2.2)	1.000	5 (13.9)	4 (8.3)	0.646
HIV		0 (0.0)	4 (8.7)	0.500	2 (5.6)	2 (4.2)	1.000
Arterial hypertension		11 (64.7)	26 (56.5)	0.766	25 (69.4)	31 (64.6)	0.815
Number of comorbidities	Median [IQR]	3 [2, 4]	3 [1, 4]	0.401	2.5 [1.8, 4.0]	3 [2, 3]	0.606
Previous CRBSI		0 (0.0)	3 (6.5)	0.679	2 (5.6)	3 (6.2)	1.000
Clinical data:							
Ward	Medical area	15 (88.2)	36 (78.3)		33 (91.7)	39 (81.2)	
Transferred from ICU		1 (5.9)	4 (8.7)	1.000	3 (8.3)	0 (0.0)	0.149
	Surgical area	2 (11.8)	10 (21.7)	0.593	3 (8.3)	9 (18.8)	0.300
COVID-19-related pneumonia		0 (0.0)	5 (10.9)	0.372	5 (13.9)	6 (12.5)	1.000
Parenteral nutrition	Yes	11 (64.7)	28 (60.9)	1.000	14 (38.9)	16 (33.3)	0.767
Immunosuppressive therapy	Yes	1 (5.9)	10 (21.7)	0.272	8 (22.2)	14 (29.2)	0.641
Ongoing ABT at diagnosis	Yes	11 (64.7)	16 (34.8)	0.065	16 (45.7)	23 (47.9)	1.000
	Missing	0	0		1	0	
	Penicillins	9 (18.8)	6 (16.7)	1.000	8 (17.4)	4 (23.5)	0.849
	Cephalosporins	7 (14.6)	3 (8.3)	0.592	5 (10.9)	2 (11.8)	1.000
	Carbapenems	2 (4.2)	2 (5.6)	1.000	2 (4.3)	1 (5.9)	1.000
	Glycopeptides, Lipopeptides, Oxazolidinones	8 (16.7)	5 (13.9)	0.965	3 (6.5)	2 (11.8)	0.874
	Antifungals *	4 (8.3)	1 (2.8)	0.549	1 (2.2)	2 (11.8)	0.357
	Others	5 (10.4)	4 (11.1)	1.000	2 (4.3)	3 (17.6)	0.226
In-hospital mortality	Yes	6 (35.3)	10 (21.7)	0.440	13 (36.1)	6 (12.5)	0.021
Discharge	Yes	11 (64.7)	36 (78.3)	0.440	22 (62.9)	41 (85.4)	0.034
	Missing				1	0	
Transfer to ICU	Yes	1 (5.9)	3 (6.5)	1.000	1 (2.8)	4 (8.3)	0.549
LOS (days)	Median [IQR]	32 [26, 49]	46.5 [32.5, 66.8]	0.038	34 [19.0, 56.5]	39.5 [28.5, 65.0]	0.250
					1	0	
VAD dwell time (days)	Median [IQR]	15 [8, 17]	12 [7.0, 21.8]	0.828	8 [4.8, 13.0]	11 [5.0, 20.2]	0.121
Time to infection (days)	Median [IQR]	25 [19.0, 32.5]	19 [10.0, 24.8]	0.173	20 [11.8, 29.2]	14 [7.8, 24.0]	0.093
	Missing	11	0	11	16	0	
Microbiological data:							
Nasal swab	Negative	10 (90.9)	28 (93.3)		17 (89.5)	28 (100.0)	
	MRSA	1 (9.1)	1 (3.3)		2 (10.5)	0 (0.0)	
	MSSA	0 (0.0)	1 (3.3)	0.631	0 (0.0)	0 (0.0)	
	Missing	6	16		17	20	
Rectal swab **	Positive	5 (41.7)	14 (41.2)	1.000	9 (40.9)	9 (32.1)	0.730
	VRE	4 (33.3)	7 (20.6)		4 (18.2)	5 (17.9)	
	KPC-producing *Klebsiella pneumoniae*	0 (0.0)	1 (2.9)		2 (9.1)	0 (0.0)	
	ESBL-producing *Escherichia coli*	1 (8.3)	9 (26.5)		4 (18.2)	4 (14.3)	
	*Candida* spp.	1 (8.3)	4 (11.8)		0 (0.0)	0 (0.0)	
	CRAB	0 (0.0)	0 (0.0)		0 (0.0)	1 (3.6)	
	NDM-producing *Klebsiella pneumoniae*	0 (0.0)	0 (0.0)		0 (0.0)	1 (3.6)	
	Missing	5	12		14	20	
Blood cultures	Polimicrobial	3 (17.6)	10 (21.7)	0.995	3 (8.3)	3 (6.2)	1.000
	*Staphylococcus aureus*	1 (5.9)	5 (10.9)	0.908	0 (0.0)	1 (2.1)	1.000
	CoNS	10 (58.8)	16 (34.8)	0.152	15 (41.7)	29 (60.4)	0.138
.	*Streptococcus* spp.	1 (5.9)	0 (0.0)	0.601	3 (8.3)	0 (0.0)	0.149
	*Enterococcus* spp.	5 (29.4)	11 (23.9)	0.905	9 (25.0)	7 (14.6)	0.356
	Enterobacteriaceae	0 (0.0)	6 (13.0)	0.279	10 (27.8)	3 (6.2)	0.016
	*Pseudomonas aeruginosa*	1 (5.9)	1 (2.2)	1.000	0 (0.0)	0 (0.0)	
	*Bacillus* spp.	0 (0.0)	1 (2.2)	1.000	1 (2.8)	1 (2.1)	1.000
	*Candida* spp.	2 (11.8)	16 (34.8)	0.138	0 (0.0)	1 (2.1)	1.000
	No isolate	0 (0.0)	0 (0.0)		1 (2.8)	8 (16.7)	0.092

HIV human immunodeficiency virus, ICU intensive care unit, CoNS coagulase-negative *Staphylococci*, MRSA methicillin-resistant *Staphylococcus aureus*, MSSA methicillin-susceptible *Staphylococcus aureus*, VRE vancomycin-resistant *Enterococcus faecium/faecalis*, ESBL extended-spectrum beta-lactamase, KPC *Klebsiella pneumoniae* carbapenemase, NDM New Delhi metallo-beta-lactamase, CRAB carbapenemase-producing *Acinetobacter baumannii*, ABT antibiotic therapy, LOS length of stay, Time to infection = number of days elapsed between VAD placement and the day of CRBSI or CABSI diagnosis. * Antifungals included fluconazole and caspofungin. ** Microbiological isolates tested positive in rectal swabs included vancomycin-resistant *Enterococci* (VRE), extended beta-lactamase-producing *Escherichia coli*, carbapenemase-producing *Pseudomonas aeruginosa*, *Staphylococcus aureus*, carbapenemase-producing *Acinetobacter baumannii*, *Candida* spp., and New Delhi metallo-beta-lactamase-producing *Klebsiella pneumoniae*.

**Table 2 microorganisms-13-01085-t002:** Epidemiological characteristics and VAD management of CRBSI and CABSI events.

		CRBSI (N = 72)	CABSI (N = 92)	All (N = 164)	*p*
VAD characteristics:	PICC	39 (54.2)	21 (22.8)	60 (36.6)	0.001
	CICC	2 (2.8)	4 (4.3)	6 (3.7)	
	FICC	5 (6.9)	6 (6.5)	11 (6.7)	
	Midline	26 (36.1)	61 (66.4)	87 (53.0)	
Median lumen number	Median [IQR]	1 [1, 2]	1 [1, 1]	1 [1, 1]	0.001
Site of insertion	Thigh	30 (41.7)	48 (52.2)	78 (47.6)	0.376
	Neck	2 (2.8)	3 (3.3)	5 (3.0)	
	Arm	40 (55.6)	41 (44.6)	81 (49.4)	
Number of attempts	Median [IQR]	1 [1, 1]	1 [1, 1]	1 [1, 1]	0.659
Microbiological isolates (BCs) *:					
Polymicrobial infection		16 (22.2)	9 (9.8)	25 (15.2)	0.047
*Staphylococcus aureus*		6 (8.3)	2 (2.2)	8 (4.9)	0.146
CoNS		34 (47.2)	49 (53.3)	83 (50.6)	0.541
*Streptococcus* spp.		1 (1.4)	3 (3.3)	4 (2.4)	0.793
*Enterococcus* spp.		18 (25.0)	17 (18.5)	35 (21.3)	0.412
Enterobacteriaceae		6 (8.3)	16 (17.4)	22 (13.4)	0.144
*Pseudomonas aeruginosa*		2 (2.8)	0 (0.0)	2 (1.2)	0.372
*Bacillus* spp.		1 (1.4)	2 (2.2)	3 (1.8)	1
*Candida* spp.		21 (29.2)	1 (1.1)	22 (13.4)	<0.0001
No microbiological isolate		0 (0.0)	10 (10.9)	10 (6.1)	0.010
Clinical decision:					
VAD removal, no re-positioning		17 (23.6)	23 (25.0)	40 (24.4)	0.982
VAD removal, re-positioned > 48 h after		10 (13.9)	5 (5.4)	15 (9.1)	0.111
VAD removal, re-positioned < 48 h after		26 (36.1)	28 (30.4)	54 (32.9)	0.548
No VAD removal + guidewire exchange		0 (0.0)	2 (2.2)	2 (1.2)	0.587
No VAD removal + lock therapy		1 (1.4)	0 (0.0)	1 (0.6)	0.901
No VAD removal + lock antibiotic therapy		5 (6.9)	1 (1.1)	6 (3.7)	0.117
No VAD removal + systemic antibiotic therapy		11 (15.3)	32 (34.8)	43 (26.2)	0.008
No VAD removal, no antibiotic therapy or lock therapy		2 (2.8)	1 (1.1)	3 (1.8)	0.829

CRBSI catheter-related bloodstream infection, CABSI catheter-associated bloodstream infection, VAD vascular access device; CoNS coagulase-negative *Staphylococci*, BCs blood cultures. * Isolates refer to blood cultures: simultaneous VAD and peripheral samples in CRBSI; single-site cultures permitted in CABSI when both were not available.

**Table 3 microorganisms-13-01085-t003:** BCs at 72 h and persistence of BSI.

		CRBSI				CABSI				TOT		
	Retention	Removal	Total	*p*	Retention	Removal	Total	*p*	Retention	Removal	Total	*p*
BCs repeated at 72 h	5 (29.4)	18 (39.1)	23 (36.5)	0.677	15 (31.2)	4 (11.1)	19 (22.6)	0.054	9 (17.0)	33 (35.1)	42 (28.6)	0.031
BSI persistence at 72 h	0 (0.0)	4 (22.2)	4 (17.4)	0.622	0 (0.0)	2 (13.3)	2 (10.5)	1	0 (0.0)	6 (18.2)	6 (14.3)	0.398

BCs blood cultures, BSI bloodstream infection.

**Table 4 microorganisms-13-01085-t004:** Multivariable mixed effect logistic regression to assess variables associated with propensity to remove catheter.

	OR	OR (95% CI)	*p*
Age	0.96	0.93; 1.00	0.071
Biological sex	0.40	0.13; 1.24	0.111
Comorbidity count	1.31	0.85; 2.02	0.212
CRBSI	1.26	0.43; 3.74	0.671
Positive rectal swab *	0.60	0.20; 1.80	0.362
Polymicrobial infection **	1.99	0.41; 9.64	0.391
CoNS **	1.15	0.40; 3.28	0.799
*Candida* spp. **	13.66	1.14; 163.76	0.039

CRBSI catheter-related bloodstream infection, CoNS coagulase-negative *Staphylococci*. * Microbiological isolates tested positive in rectal swabs included vancomycin-resistant *Enterococci* (VRE), extended beta-lactamase-producing *Escherichia coli*, *Pseudomonas aeruginosa*, *Staphylococcus aureus*, *Acinetobacter baumannii*, *Candida* spp., and New Delhi metallo-beta-lactamase-producing *Klebsiella pneumoniae*. ** Polymicrobial, CoNS and *Candida* spp. isolates refers to blood cultures.

## Data Availability

The original contributions presented in this study are included in the article/Appendix A. Further inquiries can be directed to the corresponding author.

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
