# Peer review of "Effectiveness of Vascular Catheter Removal Versus Retention in Non-ICU Patients with CRBSI or CABSI in Retrospective, Single-Center Study"

_microorganisms, 2025, doi:10.3390/microorganisms13051085_

Round 1
Reviewer 1 Report
Comments and Suggestions for Authors
- The major flaws of the data presentation and statistical methods were that the authors should present infection rates (infection episodes/catheter-day) instead of the risk ratios among the different catheters.
- The different devices/catheters may have different infection risks in their fundamental characteristics. And the sites of insertion and underlying conditions also influence the infection risks. I suggest the authors not only present overall results. You should present the results of different catheters, respectively.
Author Response
We sincerely thank the editor and the reviewers of Microorganisms for taking valuable time to thoroughly review our work. We strongly believe that a rigorous peer-review process helps us grow as both clinicians and researchers. With this in mind, we have dedicated several intensive working days to carefully addressing each and every comment, which we greatly appreciate.
Below, we provide a point-by-point response to the editor and reviewers, followed by the corresponding revisions in the manuscript.
Response to Reviewer 1
Comments:
“The major flaws of the data presentation and statistical methods were that the authors should present infection rates (infection episodes/catheter-day) instead of the risk ratios among the different catheters”.
We thank the reviewer for this suggestion. We acknowledge that catheter-related infection rates are a valuable epidemiological measure in surveillance and prevention studies. In this regard, our study group is currently working on a different paper that specifically focuses on this topic; we are greatly available to share more information about our preliminary results, if you are interested. However, when specifically considering our paper, such analysis would not directly contribute to answering our primary research question, since the primary aim of our study was not to assess the incidence of catheter-related infections per type of vascular access device, but rather to evaluate the clinical management and outcomes of patients with already established CRBSI or CABSI, in relation to catheter removal or retention. For this reason, we did not consider infection rates as a meaningful outcome in this context.
“The different devices/catheters may have different infection risks in their fundamental characteristics. And the sites of insertion and underlying conditions also influence the infection risks. I suggest the authors not only present overall results. You should present the results of different catheters, respectively”.
We agree that the type of catheter, insertion site, and patients’ underlying conditions may influence the infection risk. In response to this suggestion, we have included a table that presents detailed results stratified by catheter type and briefly described it in the results section (page 7, line 5). The table can be found in the Supplementary Materials section as “Table S1”.
Reviewer 2 Report
Comments and Suggestions for Authors
Giovanni et al. have conducted a valuable retrospective analysis examining outcomes associated with catheter retention versus removal in non-ICU patients with CABSI and CRBSI. Their finding that catheter removal may be associated with improved outcomes (mortality of 17% vs 35.8% for retention) is clinically significant, particularly the observation regarding Candida infections.
However, I have several suggestions to strengthen the manuscript:
Comment 1 (Follow-up Blood Cultures): Only 28.6% of patients had a 72-hour blood culture review, which is a critical step in managing CRBSI/CABSI. This low percentage may impact your assessment of BSI persistence. How do you view this gap in clinical practice and its potential effect on your conclusions?
Comment 2 (Differential Mortality Patterns): Why do the CABSI and CRBSI groups show different patterns regarding the impact of catheter removal on mortality rates? Are there specific physiological mechanisms or microbiological differences that could explain this observation?
Comment 3 (Table Formatting): Table formatting should follow the standard three-line format for scientific publications to improve readability and adherence to journal standards.
Comment 4 (Selection Bias): The limitations section does not address potential treatment selection bias (such as patients with less severe conditions possibly being more likely to have their catheters removed). How do you view the impact of this potential bias on the results?
Comment 5 (Catheter Type Analysis): Considering the different types of VADs (PICC, CICC, FICC, and Midline), did you perform a stratified subgroup analysis by catheter type to evaluate the effect of removal or retention on different types of catheters?
Comment 6 (Microbiological Findings): In the study, Enterobacteriaceae were found more frequently in the retention group among CABSI patients. Does this suggest that certain specific sources of infection or clinical situations might influence catheter management decisions?
Author Response
Response to Reviewer 2
Comments
“Giovanni et al. have conducted a valuable retrospective analysis examining outcomes associated with catheter retention versus removal in non-ICU patients with CABSI and CRBSI. Their finding that catheter removal may be associated with improved outcomes (mortality of 17% vs 35.8% for retention) is clinically significant, particularly the observation regarding Candida infections”.
We sincerely thank the reviewer for their positive feedback and appreciate the recognition of the clinical significance of our findings.
“Comment 1 (Follow-up Blood Cultures): Only 28.6% of patients had a 72-hour blood culture review, which is a critical step in managing CRBSI/CABSI. This low percentage may impact your assessment of BSI persistence. How do you view this gap in clinical practice and its potential effect on your conclusions?”
We strongly agree that performing follow-up blood cultures at 72 hours is a crucial step in evaluating bloodstream infections, especially considering that positive blood cultures at 72 hours are an indication to remove the catheter. As noted in our paper, only 28.6% of patients underwent 72-hour blood cultures. This finding certainly represents a limitation of our study, as it limits our possibility to make some conclusions regarding BSI persistence rates. However, we would like to underscore this clinical gap to further mark the need for more standardized protocols and educational initiatives led by vascular access teams to ensure optimal monitoring. Following your suggestion, we now further expanded this topic in the Discussion section (page 10, line 7).
“Comment 2 (Differential Mortality Patterns): Why do the CABSI and CRBSI groups show different patterns regarding the impact of catheter removal on mortality rates? Are there specific physiological mechanisms or microbiological differences that could explain this observation?”
This is an important point and we thank the reviewer for highlighting it. Based on our data, we are unable to definitively explain why catheter removal seemed to have a greater impact on mortality among CABSI patients compared to those with CRBSI.
One possible explanation is related to the limited sample size, particularly within the subgroups analyzed, which may have affected the statistical power to detect significant differences in the CRBSI group.
Another important factor to consider concerns the isolation of Candida spp. in our study, we recorded a considerable number of Candida spp infections (19), the majority of which (16) were in the CRBSI removal group. Since Candida spp isolation is associated with more severe clinical outcomes, regardless of the decision to retain or remove the catheter (Futamura A, Koseki T, Nakai T, Muroi N, Myotoku M, Iida J, Maki H, Suzuki A, Mizutani K, Ogino H, Taniguchi Y, Higashi K, Usui M. Factors Associated With Mortality in Patients With Catheter-related Bloodstream Infection: A Multicenter Retrospective Study. In Vivo. 2024 Nov-Dec;38(6):3041-3049), this may have contributed to the increased mortality observed in the CRBSI removal group. We have now highlighted this aspect in the revised Discussion (page 9, line 10).
“Comment 3 (Table Formatting): Table formatting should follow the standard three-line format for scientific publications to improve readability and adherence to journal standards”.
We thank the reviewer for this suggestion.
We have revised the formatting of all tables to follow the standard three-line format, as required by the journal’s guidelines in order to improve readability.
“Comment 4 (Selection Bias): The limitations section does not address potential treatment selection bias (such as patients with less severe conditions possibly being more likely to have their catheters removed). How do you view the impact of this potential bias on the results?”
We agree with the reviewer on the importance of acknowledging this limitation. Our study may indeed be affected by confounding by indication, as the decision to remove or retain the catheter made by the clinician could have been influenced by unmeasured clinical factors or by the perceived severity of the patient’s condition at the specific time of the infection. To partially address this issue, we included a comorbidity count as an indicator of baseline health status in our multivariable logistic regression model assessing the propensity for catheter removal. Nonetheless, we recognize that residual confounding remains possible, and we have explicitly added this as a limitation in the revised manuscript (page 10, line 18).
“Comment 5 (Catheter Type Analysis): Considering the different types of VADs (PICC, CICC, FICC, and Midline), did you perform a stratified subgroup analysis by catheter type to evaluate the effect of removal or retention on different types of catheters?”
We agree that the effect of catheter removal or retention may vary depending on the type of vascular access device. A stratified subgroup analysis by catheter type would certainly be of great interest for us. However, due to the limited sample size within each catheter subgroup, we couldn’t perform such analyses, as it would lack the statistical power to provide reliable or meaningful comparisons. To better view this topic, we added a table presenting detailed results stratified by catheter type in the “supplementary materials” section (Table S1), and briefly described it in the result section (page 7, line 5). We have also acknowledged this as a limitation in the revised manuscript, and we agree that future studies with a larger population should explore this aspect in more detail.
“Comment 6 (Microbiological Findings): In the study, Enterobacteriaceae were found more frequently in the retention group among CABSI patients. Does this suggest that certain specific sources of infection or clinical situations might influence catheter management decisions?”
We thank the reviewer for this clinically relevant observation.
As noted, Enterobacteriaceae were more frequently isolated among CABSI patients whose catheters were retained. A possible explanation is that, at the time of infection, the treating physicians may not have been fully convinced that the catheter was the primary source of the bloodstream infection, as Enterobacteriaceae are frequent cause of urinary tract infections, and consequently felt less compelled to remove the catheter. We apologize if this interpretation was not clearly presented in the original version of the Discussion. We have now revised the text to explicitly address this point and provide a clearer explanation (page 9, line 51).
Reviewer 3 Report
Comments and Suggestions for Authors
The original article concerns incidence and factors of bloodstream infections (BSI), associated with vascular access devices (catheter-associate infections). The issue of infected catheters as a life-threatening risk factor has been definitely proven elsewhere, especially in critical care patients. Therefore, catheter removal is recommended in a number of infectious complications. The authors addressed efficiency of current practice of removing/replacing VAD in these cases. A retrospective single-center study covered 2.5 year-period of observations among non-ICU patients with catheter-associated or catheter-related BSI. Higher death risk was noted for the patients in whom catheters were retained. Lethal outcomes were associated with Candida spp findings. It was concluded that VAD removal may be associated with better outcomes. The study is of some interest but has some limitations (e.g., small number and missing parameters in some subgroups)
Material and Methods: Page 3, line 4: Please specify if short-term or short- length VADs were excluded from study?
Page 3, Section 2.4. Clear difference between CABSI and CLABSI should be briefly described.
Page 3, Section 2.5. Available data: one should briefly describe the main contents of bacteriology diagnostics, its standardization, expecially, testing of antibiotic resistance for different biological materials.
Results: Pages 4-5, Table 1 should be better arranged, to be more readable. Generally, the tables in any article must contain all basic information on the patients: (1) Note median age (years), (2) ‘VAD dwell time’ and ‘Time to infection’ (probably, days). Do you mean time period after VAD installation?
Same table, rectal swabs: How many rectal samples were positive for E.coli and Klebsiella, the typical bacteria found at this site. Have You evaluated some relations between these findings and catheter-associated infections?
When listing the species and incidence of positive cultures in the tables, one should clearly specify the sources of cultures (blood, catheter, rectal swabs etc.). Comparison of detection rates at various sites for similar species could give additional information on the sensitivity of bacteriological tests, e.g., blood cultures.
Page 6, Table 2 (Microbiological characteristics) What results are provided here: positive microbial cultures from catheter, or from venous blood?
Table 3: BSI persistence at 72 hours showed no difference between patients with VAD retention or removal: the number of cases (6) is too small for this conclusion.
Page 9, Table 4: A marginally increased risk was shown for Candida infection. What types of samples were selected for this important finding: blood cultures, catheter cultures, or other studied specimens. At what time period (since catheter removal?) could the Candida positivity of predictive value?
In Methods and Results, no clear information is found about duration of observation period of the patients until lethal outcomes. The immediate causes of death (infectious complications, or underlying diseases) should be provided due to quite heterogenous cohort of patients under study.
A number of misprints are seen in the text. For example, in Abstract, line 10-11 …retained… and …retention… (may be, removal?) of catheters2.4 (Definitions) para 2, line 1. Please decipher CLABSI.
Sufficient copy-editing is necessary to make the Results more understandable.
Comments on the Quality of English LanguageIn general, the manuscript requires sufficient copy-editing.
Author Response
Response to Reviewer 3
Comments
“The original article concerns incidence and factors of bloodstream infections (BSI), associated with vascular access devices (catheter-associate infections). The issue of infected catheters as a life-threatening risk factor has been definitely proven elsewhere, especially in critical care patients. Therefore, catheter removal is recommended in a number of infectious complications. The authors addressed efficiency of current practice of removing/replacing VAD in these cases. A retrospective single-center study covered 2.5 year-period of observations among non-ICU patients with catheter-associated or catheter-related BSI. Higher death risk was noted for the patients in whom catheters were retained. Lethal outcomes were associated with Candida spp findings. It was concluded that VAD removal may be associated with better outcomes. The study is of some interest but has some limitations (e.g., small number and missing parameters in some subgroups)”
We thank the reviewer for their accurate summary and for recognizing the interest of our study.
We acknowledge the limitations related to sample size and missing data in some subgroups, as correctly pointed out. Nevertheless, we believe that our study adds valuable insight into the management of catheter-related infections in non-ICU patients, a population often underrepresented in existing literature.
“Material and Methods: Page 3, line 4: Please specify if short-term or short- length VADs were excluded from study?”
We agree with the reviewer about the need for better clarification on our methods.
To avoid any ambiguity, we have now revised the sentence to explicitly state that catheters under 15 centimeters of length (i.e. short peripheral cannulas and long peripheral catheters) were not included in the analyses (page 3, line 11).
“Page 3, Section 2.4. Clear difference between CABSI and CLABSI should be briefly described.”
We thank the reviewer for suggesting improvements in clarity for readers. We have now added a brief clarification to clearly differentiate CABSI from CLABSI as defined by the Infusion Nursing Society Standards of Practice (Nickel B, Gorski L, Kleidon T, et al. Infusion Therapy Standards of Practice, 9th Edition. J Infus Nurs. 2024) (Page 3, line 34)
“Page 3, Section 2.5. Available data: one should briefly describe the main contents of bacteriology diagnostics, its standardization, especially, testing of antibiotic resistance for different biological materials.”
We agree with the reviewer that a more detailed description of the methods would be beneficial.
We have now revised Section 2.5 to describe the microbiological methods performed during the study period. This includes the types of biological materials analyzed and standardization of microbiological procedures (page 3, line 45).
“Results: Pages 4-5, Table 1 should be better arranged, to be more readable. Generally, the tables in any article must contain all basic information on the patients: (1) Note median age (years), (2) ‘VAD dwell time’ and ‘Time to infection’ (probably, days). Do you mean time period after VAD installation?”
We thank the reviewer for pointing out these omissions.
We have improved the formatting of Table 1 to enhance readability, in accordance with scientific publication standards. Additionally, we clarified the units of measurement for ‘VAD dwell time’ and ‘Time to infection’, both expressed in days. In Table 1 notes, we also specified that ‘Time to infection’ refers to the number of days between VAD placement and the day of CRBSI or CABSI diagnosis.
“Same table, rectal swabs: How many rectal samples were positive for E.coli and Klebsiella, the typical bacteria found at this site. Have You evaluated some relations between these findings and catheter-associated infections?”
We thank the reviewer for this relevant observation.
We confirm that we collected complete data regarding rectal swab and blood cultures susceptibility testing. However, in our cohort only 21 patients had Gram-negative CABSI or CRBSI, and most episodes were sustained by susceptible strains. On the other hand, the rectal swabs performed at our center only identify ESBL and carbapenemase-producing Enterobacteriaceae, and carbapenem resistant Pseudomonas aeruginosa and Acinetobacter baumannii. In the light of these findings we considered the sample size too small to perform this potentially very interesting analysis. We acknowledge the clinical relevance of this aspect and have added more specific rectal swab isolates in Table 1 for description purposes.
“When listing the species and incidence of positive cultures in the tables, one should clearly specify the sources of cultures (blood, catheter, rectal swabs etc.). Comparison of detection rates at various sites for similar species could give additional information on the sensitivity of bacteriological tests, e.g., blood cultures”.
We thank the reviewer for this technically important suggestion.
In response, we have revised the tables to clearly indicate the source of each culture (blood, rectal swab). Although our dataset includes some cases with the same pathogen isolated from different sites (e.g., blood and rectal swabs), a formal analysis comparing detection rates across sources was beyond the scope of this study and was limited by the small sample number.
“Page 6, Table 2 (Microbiological characteristics) What results are provided here: positive microbial cultures from catheter, or from venous blood?”
We acknowledge the need for better clarification.
All microbiological isolates presented in Table 2 refer to blood cultures. According to the definitions used for CRBSI and CABSI, in the CRBSI group, isolates are always derived from simultaneous blood cultures drawn from both the vascular access device (VAD) and a peripheral vein. In the CABSI group, isolates may originate from blood cultures drawn from either the VAD or a peripheral vein, in cases where simultaneous samples from both sites were not obtainable. We have revised Table 2 and added a clarifying note to ensure this is explicitly stated in the manuscript.
“Table 3: BSI persistence at 72 hours showed no difference between patients with VAD retention or removal: the number of cases (6) is too small for this conclusion”.
We agree that the number of patients with persistent bloodstream infection at 72 hours (n=6) is too small to support any statistically meaningful comparison between the VAD retention and removal groups. We have now clarified in the results and discussion sections that this finding is purely descriptive and no firm conclusions were drawn from this subset due to the limited sample size (page 8, line 15, page 10, line 7).
“Page 9, Table 4: A marginally increased risk was shown for Candida infection. What types of samples were selected for this important finding: blood cultures, catheter cultures, or other studied specimens. At what time period (since catheter removal?) could the Candida positivity of predictive value?”
We thank the reviewer for this comment and the opportunity to clarify.
The purpose of Table 4 was to present the multivariable logistic regression analysis to evaluate factors associated with the likelihood of catheter removal at the time of infection, not with mortality. The presence of Candida spp. in blood cultures was one of the independent variables assessed in this model, as it may influence the clinical decision to remove the catheter, in line with current guidelines.
We apologize if this purpose was not sufficiently clear and have revised table 4 title and description to explicitly state the aim of this analysis.
“In Methods and Results, no clear information is found about duration of observation period of the patients until lethal outcomes. The immediate causes of death (infectious complications, or underlying diseases) should be provided due to quite heterogenous cohort of patients under study”.
We thank the reviewer for this important observation.
All patients were followed up until hospital discharge, transfer to ICU or in-hospital death, and this has now been explicitly stated in the methods section (page 3, line 17). Regarding the causes of death, we only assessed all-cause in-hospital mortality, as assessing detailed causes of mortality was beyond our study interests (page 4, line 15). In this regard, our study group is currently working on a different paper that specifically focuses on this topic; we are greatly available to share more information about our preliminary results, if you are interested. Still, we acknowledge that the population was heterogeneous and that not all deaths can be directly attributable to the catheter-related or catheter-associated infection, and we added this to the study limitations (page 10, line 22).
“A number of misprints are seen in the text. For example, in Abstract, line 10-11 …retained… and …retention… (may be, removal?) of catheters 2.4 (Definitions) para 2, line 1. Please decipher CLABSI”.
“Sufficient copy-editing is necessary to make the Results more understandable”.
We thank the reviewer for carefully pointing out these textual issues. We have corrected the misprints and clarified the acronym CLABSI as Central Line-Associated Bloodstream Infection at its first mention.
We hope that the updated and expanded version of the results will be more satisfactory.
Round 2
Reviewer 1 Report
Comments and Suggestions for Authors
The manuscript was improved after revision according to the recommendations from reviewers. I think the manuscript could be accepted for publication.
Author Response
Thank you for your comments that helped us improving the manuscript.
Reviewer 3 Report
Comments and Suggestions for Authors
The revised paper was subjected to sufficient corrections, according to the review’s remarks. It looks more understandable and organized.
Section 2.2: Case exclusion criteria are specified.
Section 2.2 (bottom). CABSI and CLABSI are clearly determined.
Section 2.5. Clinical bacteriology methods are well described.
Results: Table 1. Basic data on the patients are provided in more details, VAD persistence and terms of infection are presented as well as % of positive cultures from rectal swabs
Table 2 (Microbiological characteristics) now contains more clear information on the bacteria-positive blood cultures?
Table 4 shows a trend for age and Candida effects associated with catheter removal. A marginally increased risk was shown for Candida infection referred to blood cultures.
Limitations of the study are clearly presented at the end of Discussion.
In sum, the questions from review are answered, and one may publish this article.
Comments on the Quality of English Language
Copy editing is required for the added text fragments
Author Response
Thank you for your comments that helped us improve the manuscript.